# Ferroptosis and Its Emerging Role in Pre-Eclampsia

**DOI:** 10.3390/antiox11071282

**Published:** 2022-06-28

**Authors:** Zhixian Chen, Jianfeng Gan, Mo Zhang, Yan Du, Hongbo Zhao

**Affiliations:** 1Shanghai Key Laboratory of Female Reproductive Endocrine Related Diseases, Obstetrics and Gynecology Hospital, Fudan University, Shanghai 200011, China; 21211250003@m.fudan.edu.cn (Z.C.); 20211250008@fudan.edu.cn (J.G.); 21211250021@m.fudan.edu.cn (M.Z.); 2Department of Obstetrics and Gynecology of Shanghai Medical School, Fudan University, Shanghai 200032, China

**Keywords:** ferroptosis, pre-eclampsia, iron, lipid peroxidation, ferroptotic regulators

## Abstract

Iron is essential for cell survival, and iron deficiency is a known risk factor for many reproductive diseases. Paradoxically, such disorders are also more common in cases of iron overload. Here, we evaluated the role of ferroptosis in women’s health, particularly focusing on pre-eclampsia (PE). PE is a multisystem disorder and is one of the leading causes of maternal and perinatal morbidity and mortality, especially when the condition is of early onset. Nevertheless, the exact etiological mechanism of PE remains unclear. Interestingly, ferroptosis, as a regulated iron-dependent cell death pathway, involves a lethal accumulation of lipid peroxides and shares some characteristics with PE pathophysiology. In this review, we comprehensively reviewed and summarized recent studies investigating the molecular mechanisms involved in the regulation and execution of ferroptosis, as well as ferroptosis mechanisms in the pathology of PE. We propose that ferroptosis not only plays an important role in PE, but may also become a novel therapeutic target for PE.

## 1. Introduction

Pre-eclampsia (PE) is a major complication of pregnancy and is a significant burden to maternal and fetal health. PE is a major risk factor of premature birth and long-term cardiovascular disease (CVD) in mothers [1]. PE affects about 3–10% of all pregnancies worldwide, and together with eclampsia, they cause approximately more than 50,000 maternal deaths worldwide each year [2,3,4,5]. The Task Force of the American Congress of Obstetricians and Gynecologists (ACOG) has divided gestational hypertension into the following four groups: pre-eclampsia/eclampsia, chronic hypertension, chronic hypertension with superimposed pre-eclampsia, and gestational hypertension [6]. After 20 weeks of pregnancy, the criteria for the definition of PE include new onset hypertension, proteinuria, or damage to other end organs; eclampsia, on the other hand, is defined as grand mal seizures in women with PE [7].

PE can be classified as either early-onset or “placental” pre-eclampsia (EOPE), which appears before 34 weeks, or late-onset or “maternal” pre-eclampsia (LOPE), which appears after 34 weeks [8,9]. There are variations in epidemiology, clinical manifestations, and morbidity between EOPE and LOPE. For example, EOPE is related to a higher venture of fetal intrauterine growth restriction, while large-for-gestational-age newborns and maternal obesity are frequently linked to LOPE [10]. Globally, LOPE accounts for the majority of PE cases (80–95%) [11] and is a serious disease associated with high incidences of two life-threatening complications: eclampsia and HELLP (hemolysis, elevated liver enzymes, and thrombocytopenia) syndrome [12]. EOPE, while less common than LOPE, is associated with higher neonatal mortality and maternal morbidity [13]. Therefore, more researchers are interested in EOPE and are focusing on exploring the mechanisms behind the phenotype of this disease, which has also led to the implementation of preventive treatments (such as aspirin) and predictive biomarkers for EOPE [11].

Influencing factors of PE include family history, a genetic predisposition, race such as African heritage [14], sex cohabitation duration, maternal smoking, gravida times, maternal age, in vitro fertilization (IVF) use, and maternal health status such as obesity, hypertension, diabetes, and chronic kidney disease (CKD) [15,16,17]. An elevated risk of PE is also linked to conditions that increase placental bulk, such as multifetal pregnancies and hydatidiform mole [18]. In addition, trisomy 13 also points to a higher risk of PE [19]. Heritability of PE is estimated to be −55%, with both maternal and fetal genes contributing to risk (30–35% and 20%, respectively) [20]. Additionally, a genome-wide association study reveals that PE risk may be caused by changes close to the human fetal genome’s FLT1 locus, which is involved in the fms-like tyrosine kinase 1 (FLT1) gene [21]. Although endothelial dysfunction, unsuitable angiogenesis, insufficient trophoblast cell infiltration, and modification of the uterine spiral artery are all thought to be important causes of PE [13,22], the exact etiology of PE is still unknown.

The current management of PE includes preconception counseling, perinatal blood pressure control, complication management, timely delivery of the fetus, and postpartum monitoring [23]. Although various approaches, such as calcium supplementation, antioxidant therapy, and antiplatelet therapy, have been implemented in an attempt to reduce PE risk in susceptible women [24], the only confirmed therapy for PE is still delivery of the fetus. Expectant treatment is recommended for patients with PE who are at less than 37 weeks of gestation and have no severe symptoms, while delivery is recommended after 37 weeks. 

Ferroptosis is a recently discovered type of iron-dependent cell death (Figure 1) [25]. Ferroptosis, together with apoptosis, necroptosis, pyroptosis, netosis, and entosis, forms a cascade of different types of programmed cell death to regulate many physiological and pathological processes [26]. Ferroptosis is brought on by an overproduction of certain hydroxy-peroxidized phospholipids (Hp-PL) or by a lack of metabolic reduction ability to convert Hp-PL into non-destructive forms of phospholipids. Hp-PL buildup results in a cascade of signals that are different from other types of cell death in terms of morphology, biochemistry, and metabolic processes [27]. Although there have been many studies on the pathogenesis of PE, no definite conclusions have been reached. At present, evidence has suggested that ferroptosis may play a crucial role in placental dysfunction, thus contributing to the pathogenesis of PE [28]. The placenta is usually the primary target organ for hypoxia-reoxygenation conversions during the first trimester of pregnancy, as well as contractions of the uterus before and during labor [29,30]. Moreover, iron is abundant in placental trophoblasts due to its active transfer across the placenta to the developing fetus [31]. Some studies have recorded the lipid peroxidation of trophoblasts in placental injury [32]. Additionally, it has been discovered that human placental malfunction and PE are linked to reduced levels of glutathione peroxidase 4 (GPX4), an essential enzyme that shields cells against damaging Hp-PL species buildup and ferroptosis (see below) [33]. Interestingly, recent studies have shown that ferroptosis contributes to clinically significant placental malfunction and trophoblast damage [34]. Therefore, ferroptosis has become a very promising breakthrough point for further elucidation of the pathogenesis of PE as well as the design of new treatment methods and prevention measures.

In this review, we describe the research progress on ferroptosis and comprehensively summarize the basic characteristics, biochemical pathways, and regulatory axis of ferroptosis as well as its emerging roles in PE. We also present up-to-date evidence for the involvement of this newly defined cell death in PE pathology and treatment. Having a deeper knowledge of how ferroptosis contributes to adverse pregnancy outcomes such as PE can help with therapy, and will aid in developing a range of ferroptosis-based therapeutics for the clinic.

## 2. Ferroptosis

### 2.1. Characteristics of Ferroptosis

Cell death is essential for normal cell proliferation, growth, and development, the maintenance of homeostasis, and the prevention of hyperproliferative diseases such as cancer. The notion that the majority of programmed cell death in mammalian cells results from the activation of caspase-dependent apoptosis has been debunked by the identification of a number of non-apoptotic cell death pathways that are active at various stages of disease [35,36]. Stockwell et al. [37,38] have successively identified erastin and RSL3, collectively known as RAS-selective lethal (RSL) compounds, which have selective lethal effects on oncogenic RAS mutant cell lines. This form of cell death does not exhibit the typical signs of apoptosis, such as mitochondrial cytochrome c release, caspase activation, or chromatin breakage. Conversely, such RSL-induced mortality was discovered to be connected to elevated intracellular reactive oxygen species (ROS) levels and could be prevented by iron chelating agents or the genetic inhibition of cellular iron uptake [38,39]. RSLs, including erastin, have the potential to trigger a fatal route that is different from apoptosis, necrosis, and other well-known types of regulatory cell death. Because of the important role iron plays in this process, the above-described type of cell death is called “ferroptosis”.

The concept of ferroptosis was formally introduced in 2012 as a non-apoptotic, iron-dependent form of cell death characterized by iron overload and lipid peroxidation [25]. The Nomenclature Committee on Cell Death (NCCD) added it to the regulated cell death (RCD) family in 2018 [40]. It is characterized by an iron-dependent buildup of lipid hypoperoxides to deadly levels [41,42]. The morphology of ferroptosis is small mitochondria, accompanied by a concentration of membrane density, a reduction or disappearance of mitochondrial cristae, and a rupture of the outer mitochondrial membrane [27,43]. From the perspective of biochemistry, intracellular glutathione (GSH) is depleted, glutathione peroxidase 4 (GPX4) activity is reduced, lipid peroxides cannot be broken down by the GPX4-catalyzed reduction process, and, when Fe^2+^ oxidizes lipids, it does so in a Fenton-like manner, generating high amounts of ROSs and encouraging ferroptosis [44]. In genetics, ferroptosis is a complex biological process regulated by multiple genes. There are six genes involved in the regulation of ferroptosis induction: the tetratricopeptide repeat domain 35 (*TTC35*), the gene encoding the ribosomal protein L8 (*RPL8*), the iron response element-binding protein 2 (*IREB2*), which is a key regulator of iron metabolism, the complex component C3 of ATP synthase F0 (*ATP5G3*), citrate synthase (*CS*), and acyl-CoA synthetase family member 2 (*ACSF2*). Additionally, the expression of genes known to be involved in iron absorption, metabolism, and storage, such as transferrin receptor (*TFRC*), iron sulfur cluster as-sembly enzyme (*ISCU*), ferritin heavy chain 1 (*FTH1*), and ferritin light chain (*FTL*), are also implicated in the control of ferroptosis [25].

Cells have evolved complex systems that take advantage of and protect against ferroptosis, an advanced vulnerability brought on by the introduction of polyunsaturated fatty acids into cellular membranes. Many biological processes affect the sensitivity of cells to ferroptosis, including iron, amino acids, polyunsaturated fatty acid metabolism, and the synthesis of coenzyme Q_10_, phospholipids, NADPH, and glutathione [42]. In addition, many molecules can positively or negatively regulate ferroptosis, which includes p53, acyl-CoA synthase long-chain family 4 (ACSL4), acyl-CoA synthase 4, NADPH oxidase (NOX), and nuclear factor E2-related factor 2 (Nrf2) [45,46,47].

Ferroptosis can be assessed from two main aspects according to its signature characteristics: the lipid peroxidation level and the cellular iron level. The determination of lipid peroxidation is critical for evaluating the presence of ferroptosis. The gold standard for identifying specific oxidized lipids is oxidative lipidomics [48]. Probes such as C11-BODIPY and Liperfluo are also indirect, but they are effective methods for the detection of lipid ROSs [42,49]. The typical by-products of lipid peroxidation under oxidative stress, malondialdehyde (MDA) and 4-hydroxynonenal (4-HNE), can also be analyzed to determine the amount of lipid peroxidation [50]. An iron analysis kit or the fluorescent probe Phen Green SK (PGSK) test can be used to determine the amount of iron in cells [51]. Moreover, we can detect changes in the expression of genes associated with ferroptosis, such as *ACSL4*, *GPX4*, and *FTH1*, among others mentioned above [52]. In addition, morphologically, transmission electron microscopy can be used to observe the identification of cellular morphological features specific to ferroptosis [53]. Moreover, cell activity and death can be determined using the Cell Counting Kit-8 (CCK-8) and propidium iodide staining [54].

Numerous disorders, including degenerative ones such as Alzheimer’s disease, cancer, stroke, intracerebral hemorrhage, traumatic brain injury, and ischemic-reperfusion damage, have been linked to ferroptosis in studies. Moreover, ferroptosis may also have tumor suppressive function and can be used for cancer therapy [42]. Therefore, as a newly defined kind of cell death, ferroptosis plays a crucial role in the development of diseases, and in-depth studies are needed to further elucidate the exact mechanisms.

### 2.2. Iron Metabolism

Lipid peroxidation and susceptibility to ferroptosis are primarily caused by iron intake, storage, and export. Through Fenton and Haber-Weiss reactions, which free divalent iron may catalyze to create two highly reactive oxygen species, hydroxyl radicals, and superoxide anions, iron excess can cause membrane lipid peroxidation [37]. Importantly, iron also acts as an important cofactor in the enzymatic reactions on which the accumulation of iron-dependent lipid peroxides (lipid-ROS) need. For example, a class of non-heme iron-containing enzymes called lipoxygenases (LOXs) can catalyze the oxidation of polyunsaturated fatty acids (PUFAs) [55]. 

Iron can be exchanged inside and outside the cell through a number of pathways (Figure 1a–f). After iron is absorbed in the intestine and released into the blood from duodenal cells, ferric iron (Fe^3+^) binds to transferrin (TF) for circulation and enters the cell via the membrane protein transferrin receptor 1 (TFR1). In endosomes, the ferrireductase six-transmembrane epithelial antigen of prostate 3 (STEAP3) reduces Fe^3+^ to ferrous iron (Fe^2+^). The divalent metal transporter 1 (DMT1) can then transfer Fe^2+^ from the endosome into the cytoplasm’s unstable iron pool. Cytoplasmic iron is then stored by binding to ferritin, a protein complex represented by FTL and FTH1. Similarly, intracellular iron can be partially transported outside the cell by membrane protein ferroportin (FPN), which is thought to be the only Fe^2+^ external transporter. Furthermore, hepcidin binds to FPN, and both of these are degraded intracellularly. This prevents iron from entering the circulation and restores systemic iron homoeostasis [56,57]. An alternative pathway for the cellular export of Fe^2+^ is in the form of heme with the help of feline leukemia virus subtype C cellular receptor (FLVCR) with ATP-binding cassette protein G2 (ABCG2) [58,59]. Iron is primarily used in the mitochondrion, where it is used to create heme and iron-sulfur (Fe/S) cluster prosthetic groups. Through the SLC transporter mitoferrin (SLC25A37), iron is delivered to the mitochondrion and integrated into bioactive heme [60]. Heme oxygenase (HO1) is induced when intracellular heme is degraded, and δ-aminolevulinate synthase (ALAS) controls its synthesis [61,62]. Heme is then exported to the cytosol where it is combined with proteins [63]. 

The regulation of iron metabolism involves many processes at different dimensions, from the subcellular to the organismal. The transferrin receptor TfR, discovered in the 1970s, is involved in iron utilization, recycle, and storage [64,65]. The IRE/IRP regulatory axis was discovered in the 1980s, with the iron regulatory proteins (IRPs) as regulatory centers, using iron responsive elements (IREs) to coordinate intracellular iron absorption, storage, utilization, and excretion [66]. Moreover, *HFE*, the gene mutated in hereditary hemochromatosis, discovered in 1996, acts as a switch between the sensors of holo-Tf, TfR1 and TfR2. High holo-Tf concentrations allow TfR1 to interact with TfR2 by replacing HFE [67,68]. The discovery of the iron-regulating hormone hepcidin and its target ferroportin in the early 2000s is another significant recent discovery [69,70]. Many iron-related diseases can be attributed to genetic malfunctions affecting the hepcidin–ferroportin axis. Systemic iron homeostasis is primarily maintained by the circulating peptide hormone hepcidin and its receptor ferroportin [63]. 

Additionally, ferritin is degraded by the autophagy process known as ferritinophagy, which raises the levels of labile iron and results in ferroptosis. Nuclear receptor coactivator 4 (NCOA4), a cargo receptor involved in the ferritin degradation process that is dependent on autophagy, is necessary for this process (Figure 2). Through a direct protein–protein interaction with the iron chaperone Poly rC-binding protein 1 (PCBP1), NCOA4 mediates the flow of iron into and out of ferritin [71]. To move the cargo to the phagophores, NCOA4’s C-terminus binds to the conserved surface arginine (R23) on FTH1 and subsequently enters the autophagolysosomes, releasing free iron from ferritin and promoting ferroptosis (Figure 2a) [72]. When intracellular iron levels are high, NCOA4 is ubiquitinated by homologous to E6AP carboxy terminus (HECT) and RLD domains containing E3 ubiquitin protein ligase 2 (HERC2) and degraded by proteasomes, suggesting an inverse correlation between HERC2 and the induction of ferroptosis (Figure 2b). A lack of ferritinophagy may increase IREB2 activity and subsequently act as a feedback mechanism for upregulating transcription factors [71].

### 2.3. Lipid Peroxidation Metabolism

The degree to which cells are susceptible to ferroptosis is also influenced by lipid metabolism (Figure 1g,h). Lipid peroxidation, which is one of the characteristics of ferroptosis, occurs in polyunsaturated fatty acids (PUFAs) on specific phospholipids and directly destroys the cellular membranes, resulting in ferroptosis [73]. Due to the easily removed bis-allylic hydrogen atoms that they contain, polyunsaturated fatty acids, which are necessary for ferroptosis, are vulnerable to lipid peroxidation [74]. Therefore, the quantity and distribution of polyunsaturated fatty acids impact the level of lipid peroxidation in cells and, consequently, the effectiveness of ferroptosis. Yang et al. [74] observed that arachidonic acid or other polyunsaturated fatty acid supplements made cells more susceptible to ferroptosis. Kagan et al. [75] also found that the addition of hydroperoxyl, which derivatized from polyunsaturated fatty acid PEs to cells containing inactivated GPX4 also resulted in ferroptosis. Lipid peroxidation may directly contribute to the ferroptotic execution phase. Lipid peroxides decompose into reactive derivatives, such as Michael receptors and aldehydes, which interact with proteins and nucleic acids to cause cell death [45]. 

Meanwhile, to produce the death signals of ferroptosis, these PUFAs must be converted to coenzyme-A derivatives and then inserted into phospholipids. For the creation of lipid signal media, free polyunsaturated fatty acids must first be esterified into membrane phospholipids and then oxidized to produce ferroptotic signals [75]. The major phospholipids that can undergo oxidation and cause cell death by iron overload are phosphatidylethanolamines (PEs) containing arachidonic acid (C20:4) and its prolonged product adrenal acid (C22:4) [76]. For example, ACSL4 and LPCAT3 enzymes have been reported to be engaged in the biosynthesis and remodeling of polyunsaturated fatty acid PEs in cellular membranes [76,77]. Additionally, lipoxygenases (LOXs), a class of non-heme and iron-containing proteins, can facilitate ferroptotic peroxidation [78]. Free polyunsaturated fatty acids are the preferred substrates for LOXs, which contribute to ferritin deposition. LOXs have been found to protect against erastin-induced ferroptosis. In some contexts, LOX activity can be inhibited by a number of substances that prevent ferroptosis, including flavonoids and members of the vitamin E family (tocopherols and tocotrienols) [79,80]. However, since LOXs mainly catalyze the oxidation of free PUFAs, rather than PUFAs on pholipids at the membrane, it is suggested that 15-LOX could bind to PEBP1, a Raf kinase inhibitory protein (RKIP1), to control the Raf-1 signaling pathway mediated by mitogen-activated protein kinase (MAPK) [81]. Based on this finding, Wenzel et al. [82] went on to demonstrate that phosphatidylethanolamine-binding protein 1 (PEBP1) can bind to 15-LOX in a stable complex, allowing it to act on PUFAs related to PE and produce 15-hydroperoxyeicosatetraenoic acid-phosphatidylethanolamine (15-HpETE-PE), which induces ferroptosis.

Moreover, glutathione peroxidase 4 (GPX4) has been found to be a key inhibitor of lipid peroxidation and related pathologies. It was first recognized as phospholipid hydroperoxide glutathione peroxidase because it lessens the binding of phospholipid hydroperoxide to membranes [83]. It has been found that induced GPX4 deletion led to significant LOX-12/15-dependent lipid peroxidation and cell death in vivo [84]. Therefore, by lowering lipid peroxides and ferroptosis inhibitory proteins such as GPX4 and apoptosis-inducing factor mitochondria-associated protein 2 (AIFM2), now known as ferroptosis suppressor protein 1 (FSP1), cells can be protected against ferroptosis under normal circumstances [85,86].

### 2.4. GSH-Dependent Antioxidant Pathways: System X_c_^−^ and GPX4

System X_c_^−^, a heterodimeric amino acid consisting of two subunits, SLC7A11 and SLC3A2, can function as a cystine/glutamate antagonist system (Figure 1i). It is a crucial component of the intracellular antioxidant system and is abundantly dispersed in phospholipid bilayers. Cystine and glutamate are exchanged in the same ratio of 1:1 through system X_c_^−^ [25]. While cystine is being imported, glutamate is being transferred out of the cell as glutathione (GSH) is being produced. In the presence of glutathione peroxidase, GSH lowers ROSs and reactive nitrogen (GPXs). By preventing cystine absorption, system X_c_^−^ inhibition impacts GSH production, which in turn affects GPX activity, cellular antioxidant capability, the buildup of lipid ROS, and eventually oxidative damage and ferroptosis [87]. The core functional subunit of system X_c_^−^ is xCT, which seems to be the driver of system X_c_^−^ [88]. It is reported that the p53-xCT axis is a key potential mechanism for inducing ferroptosis, and the mutant p53 can be served as an inducer of ferroptosis. The GSH pool is depleted, and ferroptosis is induced by the interaction of the xCT inhibitor and the mutant p53 reactivator APR-246 [89]. Notably, it has been demonstrated that certain substances contribute to p53-induced ferroptosis. For example, erastin, the first ferroptosis inducer identified, increased ROS accumulation and blocked the p53-xCT pathway, suggesting that a virtuous cycle involving ROS and p53 may be established [90]. In addition to erastin, common inhibitors of system X_c_^−^ also include sulfasalazine (SAS), an anti-infective drug widely used in clinical practice. By blocking xCT, SAS can lower GSH levels and enhance ROS buildup, which eventually results in ferroptosis [91]. Apart from xCT inhibition, SAS may potentially trigger a putative ferroptosis-causing pathway. For example, in breast cancer, SAS has been reported to upregulate TFRC and DMT1, two cell-surface receptors that are both critical for iron uptake and highly required for ferroptosis [92]. Moreover, sorafenib, a traditional multikinase inhibitor, which has been approved by the FDA as a first-line treatment for advanced liver cancer, can also be utilized as an inhibitor of system X_c_^−^. Similar to erastin and SAS, sorafenib can also induce ferroptosis by targeting xCT, but the underlying mechanism remains unclear [93]. Protein biosynthesis, especially transport inhibition, may be a key target of sorafenib in promoting ferroptosis [94].

Glutathione peroxidase 4 (GPX4, that is, phospholipid hydroperoxide glutathione peroxi-dase (PHGPx)) is a special intracellular antioxidant enzyme that may directly reduce phospholipid peroxide generated in cellular membranes, even larger organic peroxides, such as polyunsaturated lipids and sterols (Figure 1i) [95]. GPX4 is a key regulator of ferroptosis induced by RSL3 and erastin, which reduce the activity of GPX4 through direct binding and the indirect loss of glutathione, respectively [96]. In order to stop the chain process of lipid peroxidation, GPX4 may convert complex hydroperoxides, such as phospholipid and cholesterol hydroperoxides, into their corresponding counterparts [97]. Therefore, when the function of GPX4 is inhibited, lipid ROS formation and lipid peroxidation levels will both increase, leading to the induction of ferroptosis. Based on this theory, the powerful protective effect of lipophilic antioxidants, such as vitamin E, which inhibits GPX4 and thus the formation of lipid peroxidation, is explained. 1S,3R-RSL3 (RSL3) is a covalent small molecule inhibitor of GPX4. RSL3 can inactivate GPX4 by binding to the active site selenocysteine, and can decrease the expression level of GPX4 and promote lipid-ROS accumulation, thus inducing ferroptosis [74,98]. In addition, the small molecules FIN56 and FINO_2_ can also indirectly inhibit the expression and/or activity of GPX4 protein by retaining only partial GPX4 characteristics [99,100]. Additionally, the production of NADPH influences how susceptible cells are to ferroptosis because NADPH stimulates glutathione reductase to convert the oxidized glutathione generated by GPX4 into GSH [101].

### 2.5. GSH-Independent Antioxidant Pathways

Except for the axis noted above, recent studies have also found a GSH-independent ferroptosis-blocking pathway. Coenzyme Q_10_ (CoQ_10_) has been identified to be a key component of the mitochondrial electron transport chain, but it can also act outside the mitochondria and inhibit lipid peroxidation by capturing free radical intermediates during the process (Figure 1j). Therefore, the depletion of CoQ_10_ can increase cell sensitivity to ferroptosis [102]. For example, a newly identified chemical inducer, FIN56, appears to cause ferroptosis by consuming both the GPX4 protein and CoQ_10_ generated from the mevalonate pathway. Statins also sensitize cells to ferroptosis by blocking the enzyme HMG-CoA reductase and inhibiting mevalonate-derived CoQ_10_ [103]. Screening for genes that inhibit ferroptosis in the absence of GPX4 revealed that FSP1 inhibits lipid peroxidation and reproduces reduced CoQ_10_, thereby inhibiting ferroptosis, suggesting that this newly identified NADPH-FSP1-CoQ_10_ pathway exerts a ferroptosis inhibitory effect in parallel with the glutathione-GPX4 axis [85,86]. In addition, DHODH (a flavin-dependent mitochondrial enzyme) can also reduce CoQ_10_ to its reduced form ubiquinol in mitochondria, to trap free radicals and inhibit lipid peroxidation and thus suppressing ferroptosis [104]. Another GSH-independent ferroptosis-blocking pathway is the rate-limiting step of the metabolite tetrahydrobiopterin (BH4) production that involves the GTP cyclo hydrolase 1 (*GCH1*) gene. BH4 suppresses ferroptosis by helping to form reduced CoQ_10_ and blocking specific lipid peroxidation reactions [105].

## 3. Ferroptosis and Pre-Eclampsia

### 3.1. The Role of Iron in PE Pathology 

During pregnancy, maternal iron requirements will increase by about 30% to support maternal and infant hematopoiesis, and iron loss is reduced due to, for example, cessation of menstruation during pregnancy. Therefore, pregnant women may have a decreased ability for feedback systems to react to ambient iron exposure, resulting in vulnerability to clinical diseases caused by excess iron [106]. As mentioned above, the hepcidin-ferroportin cycle is a crucial mechanism for the regulation of iron homeostasis. Hepcidin restricts iron transport through ferroportin and reduces plasma iron concentrations. Therefore, in healthy pregnancies, the expression level of hepcidin is reduced to increase the absorption of iron in the intestinal tract and the mobilization of stored iron in the liver and spleen, so as to meet the body’s increased iron requirements during pregnancy. To summarize, with the progress of pregnancy, the expression of hepcidin decreases, dietary iron absorption increases, and stored iron release increases. Since the concentrations of hepcidin are inversely connected to the transfer of iron across the placenta, iron transfer to the fetus also increases accordingly [107]. In addition, maternal levels of ferritin and transferrin-saturation decrease, while TfR levels increase during the second and third trimesters of normal pregnancy [106]. However, in PE, levels of free iron, ferritin, and transferrin-saturation increase, while TfR levels decrease [108,109,110]. However, some studies have shown a decrease in the level of hepcidin in PE patients [111], while others have found an elevated hepcidin level in early pregnancy or a normal level in late pregnancy [112]. It has been reported that an elevated serum hepcidin level in early pregnancy is associated with PE occurrence [112]. So far, current studies on hepcidin level changes in PE patients are controversial, and further studies are needed to elucidate the mechanism. 

Moreover, iron levels were significantly correlated with disease severity, and serum iron and ferritin levels in EOPE were distinctly higher than those in LOPE and control groups [113]. It has also been shown that, in 18% of PE patients, the transferrin saturation levels are high, which is related to iron overload [114]. During normal pregnancy, the maternal plasma blood volume increases at the first trimester and can increase by 30–50% at the third trimester [115]. This hypervolemia has little effect on pregnancy complications, while increased plasma volume can cause changes in iron concentration and other factors, which have different effects on normal pregnancy and PE pregnancy [116]. Therefore, the abundance of iron in trophoblasts can lead to iron overload, resulting in ferroptosis, to which human trophoblast cells are sensitive, leading to macro-blebbing and vesiculation of the plasma membrane, further resulting in placental dysfunction and trophoblast injury [117]. Total Hb levels in maternal blood also fluctuate throughout pregnancy, and both low and high levels are linked to adverse pregnancy outcomes. The likelihood of adverse birth outcomes is inversely correlated with maternal hemoglobin concentration or iron status during pregnancy. High Hb levels, however, as well as high iron levels, appear to be linked to less favorable pregnancy outcomes [106]. Similarly, in a randomized controlled trial, Ziaei et al. [118] found that women who had higher mean hemoglobin concentrations in the third trimester of pregnancy had an increased risk of hypertension (2.7% vs. 0.8%, *p* < 0.05), as well as an increased risk of giving birth to small-for-gestational-age (SGA) infants (15.7% vs. 10.3%, *p* = 0.035). 

Obesity is a risk factor of PE and has been relevant to increased inflammation due to elevated levels of interleukin-6 (IL-6), which is a vital regulator of hepcidin and thus may be associated with iron dysregulation [119]. Women with PE showed a marked inflammatory response and significantly increased levels of IL-6 and tumor necrosis factor-α (TNF-α) [120]. Race is another influencing factor of PE. Mutations in the regulatory gene *HFE*, which regulates iron homeostasis in vivo, are involved in the regulation of hepcidin expression, leading to hemochromatosis or excessive iron storage, with the two most common mutations (C282Y and H63D) occurring in 14% and 29% of the European White populations, respectively [121]. In addition, it has been reported that a gene mutation in ferroportin (Q248H) leads to partial resistance to hepcidin-induced differentiation, with a prevalence of up to 13% in Africans [122]. To support the normal increase of erythropoiesis in late pregnancy, the first and third trimesters see nearly a doubling of EPO levels [106]. However, women with PE had lower levels of EPO at the third trimester compared to normal pregnancies [111]. Interestingly, although total serum transferrin levels of PE patients increased during pregnancy compared with non-pregnancy, they were still much lower than normal pregnancy [123]. Moreover, iron supplementation during pregnancy in iron-deficient women may inhibit maternal hepcidin production and/or activity [124], causing a continuation of dietary iron absorption, an increase in hemoglobin levels, a rise in blood viscosity, and finally a reduction in placental blood flow [118]. Furthermore, excessive dietary iron intake may cause elevated circulating levels of postprandial non-transferrin bound iron (NTBI), resulting in oxidative stress, lipid peroxidation, and the DNA damage of placental cells, which affect placental function [125]. It has also been found that miR-30b-5p in PE models can reduce the expression of ferroportin 1 by downregulating Cys2/glutamate antiporter and PAX3, leading to a decrease in GSH and an increase in labile Fe^2+^, therefore promoting the occurrence of ferroptosis in PE patients [126]. The expression of miR-210 was also found to be increased in placenta in PE models, resulting in iron accumulation and autophagosome formation in trophoblast cells and hemosiderin deposition in placental stroma trophoblasts [127]. Genetic studies have also found that, in PE, the expression levels of genes mainly responsible for iron metabolism (*FTH1* and *FTL*) are downregulated, leading to disruption of iron uptake and intracellular storage, thereby promoting cellular ferroptosis [128].

### 3.2. The Role of Oxidative Stress in PE Pathology

Until 8–10 weeks of gestation, when the embryo is in a hypoxic and hypoglycemia environment, the maternal spiral arteries are fully obstructed by clots of endothelial cells and blood. The spiral arteries are completely canalized by 10–12 weeks of pregnancy, maternal blood rushes into the placental lacunae, and the fetal villi are exposed to glucose, oxygen, and iron for the first time. Inadequate remodeling of the maternal spiral arteries and shallow EVCT endovascular invasion, which are the pathological features of PE, can result from such rapid perfusion, which can also cause a significant amount of oxidative stress and tissue injury. These outcomes include lipid peroxidation of cell membranes and excessive ferroptosis at the maternal–fetal interface, mainly in trophoblast cells [28]. Poorly remodeled spiral arteries pose risks of placental underperfusion, high velocity, and turbulent blood flow, resulting in placental ischemia [129] and oxidative stress [130], which will cause damage to the placental villi and lead to abnormal levels of angiogenic proteins in maternal blood [131]. Excessive secretion of anti-angiogenic factors leads to vascular inflammation, endothelial dysfunction, and maternal vascular damage [130], which ultimately results in clinical manifestations of hypertension and multiple maternal organ damage. The production of anti-angiogenic factors increased, such as soluble fms-like tyrosine kinase-1 (sFlt-1) [132], a protein that binds to the functional receptor of vascular endothelial growth factor (VEGF), and soluble endoglin (sENG) [133]. However, the release of angiogenic placental growth factor (PIGF) is inhibited [134], resulting in an imbalance among them. Therefore, this two-stage paradigm of early placental dysplasia coupled with severe maternal organ damage and systemic endothelial dysfunction, first proposed in 1993 [135], could be a possible model for determining the pathogenesis of PE (Figure 3).

As the most mutagenic product of lipid peroxidation [50], as well as an oxidation product that may lead to ferroptosis [136], serum malondialdehyde (MDA) levels have been shown to be significantly elevated in PE and eclampsia patients in multiple studies [137,138,139]. In metabolomic analysis of placental mitochondria, PUFA levels and other mitochondrial abnormalities were significantly higher in patients with severe PE than in controls with normal blood pressure [140]. Additionally, histological examinations of the spiral artery walls in many PE patients reveal a buildup of “foam cells” (macrophages harboring low-density lipoproteins) that are lipid-filled, similar to the early stages of atherosclerosis. Although such abnormalities have been observed for decades, the mechanisms behind this type of acute atherosclerosis are largely unknown. One of the key factors might be ferroptosis with lipid peroxidation [141]. Moreover, Alahari et al. [142] proposed that PE is caused by chronic hypoxia and iron homeostasis disorders at the maternal–fetal interface. Gene expression levels of the von Hippel Lindau (VHL) protein, a key executor of the cellular hypoxia response in PE, are regulated by histone demethylase JMJD6 (Jumonji domain containing protein 6). In PE patients, hypoxia and Fe^2+^ bioavailability leads to decreased JMJD6 demethylase activity, resulting in downregulated VHL expression, accompanied by changes in placental morphology and reduced pup weights. Moreover, multiple experiments have shown that SOD and GSH-Px activities are significantly decreased, catalase activity is increased, and lipid peroxidation and thromboxane (TX) secretion are increased in the placenta of women with PE [143,144,145]. Vaughan et al. [146] exposed the ED_27_ trophoblast cell line to an oxidizing solution that was rich in Fe^2+^ and linoleic acid, causing the same changes as in the placenta of women with PE, and these changes were prevented by Fe^2+^ chelation in the oxidizing solution, except for TX, suggesting that oxidative stress combined with elevated maternal circulating lipids and Fe^2+^ levels may lead to placental oxidative stress, abnormal placental antioxidants, and TX in PE. The expression profile and function of ferroptosis-related genes (FRGs) in PE also showed that the *HIF1α* and *MAPK8* genes were downregulated and that the *PLIN2* gene was upregulated [128]. *HIF1α* is the main transcriptional regulator of hypoxia response and regulates cell survival during stress response. It can also reduce fatty acid β-oxidation and promote lipid storage [147,148]. The expression of hypoxia-induced genes such as erythropoietin, vascular endothelial growth factor (VEGF), and nitric oxide (NO) synthase is regulated by *HIF1α* and *HIF2α*, which are by-products of related oxygen sensing pathways. The expression of HIF1α in the human placenta increases in early gestation and decreases at around 9 weeks, when fetal circulation and oxygenation increase [149]. In PE, HIF1α and HIF2α are overexpressed in the placenta and cannot be downregulated during oxygenation [150]. Thus, HIFlα appears to be the pathogenic mediator of PE. Likewise, as a member of the mitogen-activated protein kinase (MAPK) family, MAPK8 is activated by environmental stressors and participates in the regulation of multiple signaling pathways, which plays a crucial role in many processes of cell functions [151]. Adipogenic differentiation-related protein (ADRP), also known as perisidine 2 (PLIN2), can be enclosed in lipid droplets together with phospholipids to take part in neutral lipid storage [152]. Furthermore, advanced oxidation protein products (AOPPs), as novel markers of oxidative stress, have been shown to cause trophoblast cell damage and dysfunction and play a role as a new pathogenic agent in PE. Clinical trials also showed significant differences in mean AOPP levels among normotensive pregnant women and pregnant women with severe or mild PE, and plasma AOPP level was positively correlated with 24 h proteinuria excretion and cystatin C [153]. It has also been proved that, in trophoblast cells, ROSs were shown to promote miR-335-5p expression in a p53-dependent manner, further reducing the expression of specific protein 1 (Sp1) and thus inhibiting the transition and migration of epithelial cells to mesenchymal cells [154]. The discoveries of these new targets have provided new ideas and evidence for the etiology of PE.

### 3.3. The Role of Other Ferroptosis Regulators in PE Pathology

Boutet et al. [155] found that heme oxygenase-1 (HO-1) and HSP-70 mRNA expression in whole blood was dramatically elevated during fetal and maternal circulations. Blood levels of GPX4 mRNA in the PE group were 1.6 times higher than that in the normal pregnancy group, which suggest that PE is correlated with specific antioxidant responses in maternal and fetal circulation, possibly associated with harmful oxidative stress responses observed in the syndrome. Moreover, genotyping studies also showed that there were significant statistical differences in genotype and allelic frequencies of rs713041 in *GPX4* between PE patients and the normal pregnancy group, and the C allele had a higher risk for the pathogenesis of PE. At the same time, the rs713041 genotype was also found to be associated with mild, severe, and early-onset PE. Therefore, rs713041 in *GPX4* may play a vital role in the pathogenesis of PE [33].

The nuclear factor erythroid-2-related factor 2 (Nrf2), a major regulator of antioxidant response, has recently been shown to prevent ferroptosis (Figure 1k) [156]. Wang et al. [157] found that Fe^2+^ content was upregulated and the expressions of SLC7A11, GPX4, and FPN1 were downregulated in PE patients. Hypoxia can promote the translocation of Nrf2 to the nucleus, leading to the activation of the Nrf2/HO-1 signaling pathway. Hypoxia-induced Nrf2 overexpression can inhibit the levels of GSH, MDA, ROSs, and Fe^2+^ and can promote the activation of the Nrf2/HO-1 signaling pathway and the expressions of SLC7A11, GPX4, and FPN1, which indicate that Nrf2 signaling activation could relieve hypoxia and play a protective role in PE. Interestingly, it has been found that DJ-1, as an important sensor of intracellular redox state, its mRNA expression levels were significantly increased in PE patients [158]. Its expression level is inversely connected to MDA concentration and favorably correlated with the Nrf2/GPX4 signaling pathway expression levels. In order to protect the body from toxins, DJ-1 can dissociate Nrf2 and Keap-1, translocating Nrf2 into the nucleus where it binds with antioxidant response element (ARE) [159]. This increases the expression of a series of downstream antioxidant enzymes, such as GPX4 and SOD. In other words, DJ-1 can control the Nrf2/GPX4 signaling pathway, causing ferroptosis in trophoblast cells and acting as a protective factor in the pathogenesis of PE [160]. 

CoQ_10_ is essential for energy production and the development of ROSs since it is the only non-polar electron transporter in the mitochondrial respiratory chain [161]. Circulating CoQ_10_, either directly or through the production of vitamin E, is considered as a potential antioxidant [162]. Therefore, it makes sense that CoQ_10_ might contribute to the onset of PE. Initial studies have shown that CoQ_10_ levels in PE patients rise gradually from the first trimester and continue to rise until delivery [163]. However, subsequent studies have shown that CoQ_10_ levels are significantly downregulated in women with PE [164]. A hypothesis was proposed suggesting that there might be a “mechanism” for CoQ_10_ consumption during PE (independent of diet), possibly due to an increased production of ROSs [165]. Intriguingly, the researchers discovered significantly higher amounts of CoQ_10_ in the placenta and the umbilical cord of women with PE, suggesting compensatory accumulation [166]. To control for altitude, several subsequent studies at sea level showed that CoQ_10_ levels were significantly reduced in normal pregnant women, but the difference was less pronounced in women with PE than in those living at higher altitudes. However, at sea level, CoQ_10_ levels were also significantly higher in the placentas of women with PE [167]. Subsequent clinical trials showed that CoQ_10_ supplementation was an effective intervention to reduce PE risk [168]. In patients with PE, although CoQ_10_ reduced the incidence of PE, and the levels of CoQ_10_ in placental tissues were high, the mitochondrial levels of CoQ_10_ did not change significantly [169]. In other words, CoQ_10_ levels may have varied between healthy pregnant women and PE patients. The plasma and placenta both undergo these modifications, but they are more likely to be found in mitochondria. It is unknown whether the change in the level of CoQ_10_ in PE patients involves the participation of ferroptosis mechanism, and more research are needed.

## 4. Ferroptosis and PE Therapy

Because placental hypoxia is the core of PE pathogenesis, it has been a main focus for developing novel treatments for PE. Ouabain, a digoxin-like molecule that can inhibit HIF1 and HIF2, has been shown to block sFlt-1 (a soluble splice variant of the membrane-bound receptor VEGFR1) production and reduce hypertension in placental ischemia rats [170]. In addition, antioxidants have been studied for their potential beneficial effects on PE. For example, in a clinical trial of aspirin, treatment started at ≤16 weeks of gestation at a daily dose of ≥100 mg significantly reduced the risk of preterm PE compared with patients who were not treated with aspirin (relative risk: 0.33; 95% confidence interval: 0.19–0.57) [171]. Additionally, studies have showed that aspirin treatment at 150 mg per day reduced preterm PE by 62% when compared to the placebo [172]. Low-dose aspirin is now recommended for PE prevention in high-risk women, although there is currently a dearth of conclusive evidence proving how aspirin treatment reduces prenatal morbidity and death [173]. Studies have also shown that, in high-risk women, the initiation of treatment with low-molecular-weight heparin before 16 weeks of gestation can significantly reduce the risk of PE and other placenta-mediated complications. Combined treatment with low-dose aspirin significantly reduced the risk of PE compared with low-dose aspirin alone [174,175]. However, non-specific antioxidants, such as the use of vitamin C and vitamin E, have not been shown to be effective in preventing PE in clinical trials [176,177]. As a result, the causes of oxidative stress in PE patients are receiving increasing amounts of attention in order to further identify therapeutic targets. Mitochondrial oxidative stress has become an attractive target, and the use of mitochondrial-targeted antioxidants as a therapeutic strategy to reverse oxidative stress in PE has become an emerging research topic [178]. For example, AP39, a novel mitochondrial-targeted hydrogen sulfide donor, has been reported to block ROS production, reduce HIF-1α protein levels, and reduce sFLT1 production. At the same time, as a mitochondrial bioregulatory factor, it can enhance the activity of cytochrome c oxidase and reverse the oxidative stress and anti-angiogenesis response of hypoxic trophoblast cells [179]. 

In addition, statins are effective in the long-term prevention of cardiovascular diseases and mortality, not only through their lipid-lowering mechanisms but also through their regulation of inflammation, antioxidants, and endothelial homeostasis, among other pleiotropic effects [180]. The inhibition of HMG-CoA reductase by statins disrupts membrane synthesis [181]. They may also interfere with cell proliferation, growth, and metabolism and protein glycosylation, which all play important roles in normal placental development. Statins may also disrupt the placenta by expressing peroxisome proliferator-activated receptor γ and inhibit trophoblast invasion [182]. In a patient with poor obstetric outcomes, Otten et al. [183] administered 10 mg of pravastatin daily beginning in the second trimester of pregnancy, achieving a normal pregnancy and full-term delivery of a healthy neonate suitable for gestational age in a patient with a history of severe, early-onset, recurrent HELLP syndrome. There are various findings supporting the use of pravastatin as the statin of choice for the treatment of early and severe PE. Pravastatin, as the most hydrophilic statin, can restrict placental transfer and be metabolized and cleared through liver and kidney pathways. In addition, pravastatin has shown multifactorial activities in sFlt-1-induced mouse PE, e.g., promoting the expression level of VEGF, fueling endothelial function, upregulating vascular eNOS, and stimulating vascular reactivity. The use of statins to treat PE has become dominant in the past few years [182].

However, although these agents have been shown to be associated with oxidative stress in the placenta, their specific roles in the regulation of ferroptosis remain unclear. To the best of our knowledge, currently there is no report on specific treatment of the ferroptosis regulation pathway in PE, which warrants further investigation.

At the same time, excluding oxidative stress, iron overload is another landmark feature of ferroptosis, and its treatment scheme as a target is also a hotspot in ferroptosis research. Areas of interest include the use of iron chelating agents such as desferrioxamine and deferiprone. However, maternal desferrioxamine therapy is mostly used in pregnant women with beta-thalassemia [183], and its application in PE pregnant women has not been effectively studied. Therefore, in order to find more effective treatment options for PE patients based on ferroptosis, iron chelating agents require further study.

## 5. Conclusions and Future Perspectives

Ferroptosis, a newly defined type of RCD, is driven by a buildup of iron-dependent lipid ROS and has been linked to the development of numerous illnesses. In pregnant women, ferroptosis affects the function of placenta and participates in the development of multiple adverse pregnancy disorders, such as PE, posing a serious threat to maternal and fetal health (Table 1). The common mechanisms of ferroptosis in PE include iron overload, lipid peroxide accretion, GPX4 suppression, and systemic X_c_^−^ inhibition, which are correlated with the findings of other diseases. Ferroptosis-related PE is also greatly mediated by important regulatory factors such GPX4, SLC7A11, Nrf2, and CoQ_10_. However, it is still unclear whether PE has particular signaling pathways or regulators. Therefore, more studies are needed to further clarify the role of ferroptosis in the pathogenesis of PE. As a result, determining the specific mechanisms of ferroptosis in PE and developing more effective therapeutic regimens has become a hotspot for future research. In addition, further research is also necessary to determine whether ferroptosis and other types of cell death in PE are related, as well as whether these RCDs might have similar routes and important modulators, so as to provide new directions for the combination of different therapeutic interventions.

Currently, there are various studies on the treatment of PE, and many agents based on placental oxidative stress have been proved to be effective, such as ouabain, aspirin, low-molecular-weight heparin, and mitochondrial-targeted antioxidants such as AP39 and pravastatin. However, there are still no definitive treatments for ferroptosis in PE. Therefore, for the benefit of patients, more ferroptosis-specific therapies are urgently needed. Similarly, studies on biomarkers of ferroptosis in PE need to be carried out, as it may contribute to early detection and diagnosis of the disease and may predict the severity of the disorder. In addition, studies of signaling pathways and major transcriptional regulators of ferroptosis are also needed so that we can benefit more from its modulation and thus apply them to the treatment of more kind of diseases. Therefore, as a novel therapeutic target, ferroptosis should be further investigated in the field of PE.

## Figures and Tables

**Figure 1 antioxidants-11-01282-f001:**
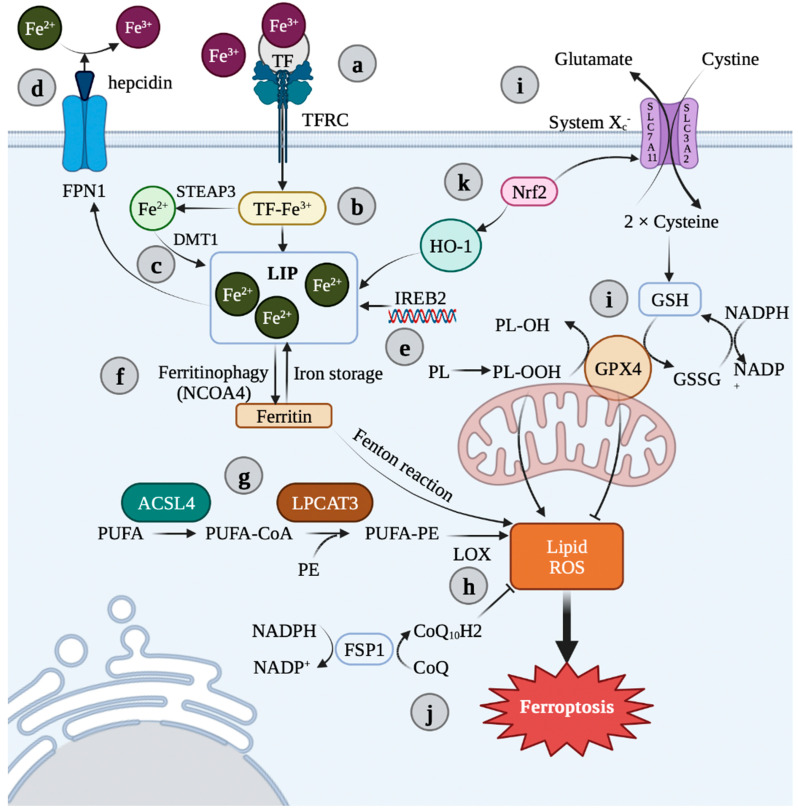
Overview of ferroptosis pathways. (**a**) TFRC is bound to TF and two Fe^3+^, which then enter the cell via endocytosis. (**b**) In endosomes, Fe^3+^ is reduced to Fe^2+^ via STEAP3. (**c**) Fe^2+^ can then be stored as ferritin or act in an active loose state known as a “liable iron pool” (LIP) by DMT1. (**d**) Fe^2+^ can be released into the plasma via ferroportin (FPN), while hepcidin can bind to FPN to inhibit Fe^2+^ release and regulate iron homeostasis. (**e**) IREB2 is also a key regulator of iron content. (**f**) Ferritinophagy and LIP contribute to iron load, and excess iron is the cofactor of LOX. (**g**) ACSL4 and LPCAT3 are involved in the manufacture and modification of PUFA-PE containing polyunsaturated fatty acid in the cell membrane. (**h**) LOX (mainly LOX-15) mediates the peroxidation of PUFA-PE to conduct the ferroptosis axis. (**i**) Ferroptosis is controlled by two major regulatory systems, namely, the transporter system X_c_^−^, which is composed of SLC3A2 and SLC7A11, and the GSH/GPX4. (**j**) In addition, there are GSH-independent antioxidant pathways, such as the CoQ_10_ axis. The FSP1 in the plasma membrane showed oxidoreductase activity. It reduces the coenzyme Q and decreases the accumulation of L-OOH. (**k**) The Nrf2/HO-1 axis can promote the increase of Fe^2+^ by catabolizing heme. Image was created with BioRender.com (accessed on 25 June 2022). FPN 1: ferroportin 1; TF: transferrin; TFRC: transferrin receptor; LIP: liable iron pool; Nrf2: nuclear erythroid 2-related factor 2; HO-1: heme oxygenase-1; SLC7A11: solute carrier gamily 7 member 11; SLC3A2: solute carrier gamily 3 member 2; IREB2: iron responsive element binding protein 2; GSH: glutathione; GSSG: oxidized glutathione; GPX4: glutathione peroxidase 4; NCOA4: nuclear receptor coactivator 4; ACSL4: Acyl-CoA synthetase long-chain family member 4; LPCAT3: lysophosphatidylcholineacyl transferase 3; PUFA-PE: phosphatidylethanolamine; LOX: lipoxygenase; ROS: reactive oxygen species; FSP1: ferroptosis suppressor protein 1; Fe^3+^: ferric iron; Fe^2+^: ferrous iron; STEAP3: six-transmembrane epithelial antigen of prostate 3; DMT1: divalent metal transporter 1.

**Figure 2 antioxidants-11-01282-f002:**
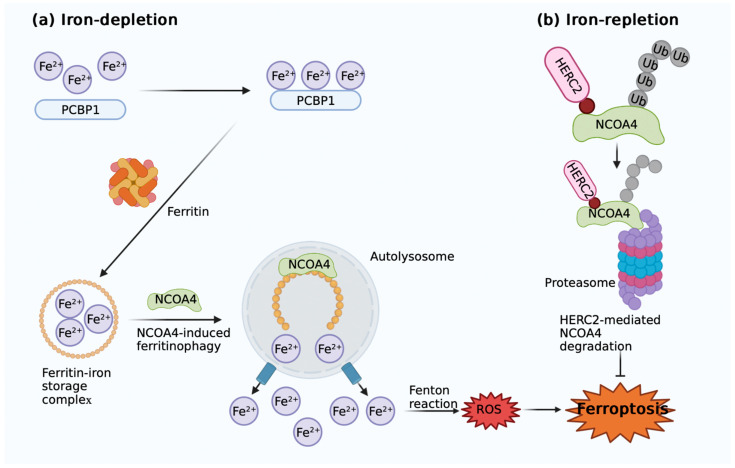
Mechanism of ferritinophagy. (**a**) NCOA4-mediated ferritinophagy promotes ferroptosis induction. NCOA4, coupled with PCBP1, binds to ferritin and mediates its degradation as well as subsequent iron release. (**b**) HERC2 mediates NCOA4 degradation. HERC2 mediates NCOA4 degradation in ubiquitin-dependent ways when iron depletion is present, thereby preventing ferritinophagy and ferroptosis. Image was created with BioRender.com (accessed on 25 June 2022). PCBP1: iron chaperones poly rC-binding protein 1; NCOA4: nuclear receptor coactivator 4; ROS: reactive oxygen species; HERC2: homologous to E6AP carboxy terminus (HECT) E3 ubiquitin protein ligase 2; Ub: ubiquitin.

**Figure 3 antioxidants-11-01282-f003:**
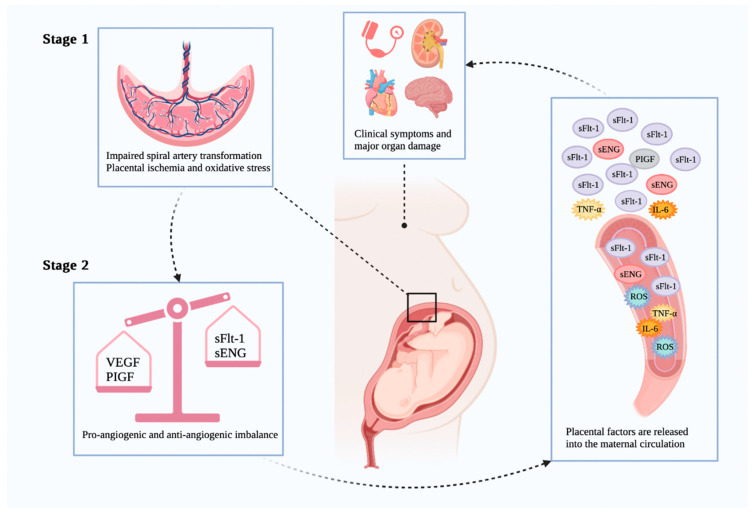
The two-stage model of PE pathogenesis. Image was created with BioRender.com (accessed on 25 June 2022). VEGF: vascular endothelial growth factor; PIGF: placental growth factor; sFlt-1: soluble fms-like tyrosine kinase-1; sENG: soluble endoglin; ROS: reactive oxygen species; IL-6: interleukin-6; TNF-α: tumor necrosis factor-α.

**Table 1 antioxidants-11-01282-t001:** Research progress on ferroptosis in PE.

Author (Ref)	Year	Related Pathway	Target	Findings
Boutet M [157]	2008	GSH synthesis pathway	GPX4	Over expression of GPX4 in PE group
Peng XG [33]	2016	GSH synthesis pathway	GPX4	rs713041 in *GPX4* gene and the C allele has the higher risk for pathogenesis of PE
Zhang H [128]	2020	GSH synthesis and iron metabolism pathway	FPN1	Downregulation of Cys2/glutamate antiporter, PAX3 and GSH levels, upregulation of labile Fe^2+^ due to miR-30b-5p
Wang Y [159]	2021	GSH synthesis and iron metabolism pathway	System X_c_^−^ GPX4 FPN1	Activation of Nrf2/HO-1 signaling pathway and overexpression of SLC7A11, GPX4 and FPN1 due to Nrf2 overexpression
Liao TT [162]	2022	GSH synthesis pathway	GPX4	Upregulation of Nrf2/GPX4 signaling pathway and resist ferroptosis in the pathogenesis of PE

PE: pre-eclampsia; GSH: glutathione; Cys2: cysteine 2; PAX3: paired box 3; FPN1: ferroportin 1; GPX4: glutathione peroxidase 4; Nrf2: nuclear erythroid 2-related factor 2; HO-1: heme oxygenase-1; SLC7A11: solute carrier gamily 7 member 11.

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
