# Peer review of "Ferroptosis and Its Emerging Role in Pre-Eclampsia"

_antioxidants, 2022, doi:10.3390/antiox11071282_

Round 1

Reviewer 1 Report

This review By Chen et al, deals with ferroptosis in preeclampsia.

The introduction is very general and well-written. It defines mild and severe preeclampsia, and the different admitted subtypes of the disease. It then lists predisposing factors. Reference 19 refers to a review. Better to quote the original research paper PMID: 15384082.

The second part of the introduction explains ferroptosis, and is nicely illustrated and summarized in Figure 1. The figure is divided into letters a,b, c, etc that are not presented in the image. It is disturbing since later figure 1f and 1g are mentioned (line 260), 1h line 310. As a curiosity from my part as a nonexpert in ferroptosis, the iron metabolism is described with several important proteins (hepcidin, ferroportin), but they are not part of the picture. It could be nice to extend the figure to append these factors that ensure the iron availability (as well as alpha1-microglobulin, etc.).

Line 104 ‘in this review, we reviewed’ is inelegant.

The second part describes ferroptosis in more detail, and is useful for the non-specialized reader, explaining the dynamics of iron in the organism. The pivotal relation between iron and lipid peroxidation is well explained, and how fatty acid modifications are achieved when iron is present.

2.4 and 2.5 describes the antioxidant GSH-dependent and independent pathways.

Part 3 put together the literature comibining ferroptosis and preeclampsia. Line 390 written ‘hapcidin’ instead of ‘hepcidin’.

The relation between the different types of preeclampsia and iron levels is reminded, together with the links with oxidative stress, which is clearly related to free iron levels.

Part 4 evokes possible therapeutic pathways, although as mentioned the involvement of specific treatments leading to decreased ferroptosis is still unclear.

The review tends to prove that ferroptosis affects placental health, and share common mechanisms with Preeclampsia, albeit the causal link is not so clear-cut.

Overall, this review dwell on a hypothesis and conclude that consistent pathways are present, warranting further scientific exploration.  The text is well written and give consistent information to the reader. I have only a concern, I could not identify specific works from the authors from the reference list. Do they have an ORCID number that would allow the reader to know the scientific contribution of the authors to the field of obstetrics or/and ferroptosis?

Reviewer 2 Report

In this review paper is reported the mechanism of ferroptosis, its relation to pre-eclampsia, and potential therapeutic methods. I think this paper should be eventually published. I have only minor comments to this paper.

1. Introduction

Ferroptosis firstly appears in line 81, and it seem that the lines 1-81 seems too long.

2. Figure 1

In the figure legend the authors describe the detailed mechanisms as a,b,.....h,i.  It would be helpful if a, b, .....h, i are indicated in the figure.

3. Ferroptosis is defined as iron-dependent cell death. Then, one treatment strategy should be to use iron chelating therapeutics such as desferruixamine or deferiprone. Please discuss on it.

4. On the other hand, anemia and iron deficiency are often seen during pregnancy....

5. Conclusion and future perspectives.

The authors lastly conclude the necessity of studies on biomarkers and therapeutic methods for ferroptosis. However, the readers would expect that the authors should be the experts on this research field, struggling with such studies and have some results or future visions on them. Please describe the recent works of the authors as researchers, not as just reviewers of literatures, with authors' reference papers.

Reviewer 3 Report

Firstly, it should be pointed out that this review has a high percentage of plagiarism (41%, if we eliminate the bibliographical references).

The authors conduct a literature review focusing on the role of ferroptosis in women's health, especially in pre-eclampsia. They review recent studies investigating the molecular mechanisms involved in the regulation and execution of ferroptosis, as well as the mechanisms of ferroptosis on pre-eclampsia. They conclude that ferroptosis not only plays an important role in PE, but could become a therapeutic tool.

I consider this to be an interesting and scientifically interesting paper.

However, I think it is excessively long as it describes in excessive detail processes associated with ferroptosis, but which are very generic, such as iron metabolism or lipoperoxidation. These sections could be summarised a little to make the reading more fluid.

It is also important to reduce the level of plagiarism observed.

Minority

Page 5, line 192: Correct the term "Carcer".

Page 9, line 390: Correct "Hapcidin".
